# Rubber Rail Pad Reinforced by Modified Silica Using GPTMS and Sulfenamide Accelerator

**DOI:** 10.3390/polym14091767

**Published:** 2022-04-27

**Authors:** Rudeerat Suntako

**Affiliations:** Department of Physics, Faculty of Liberal Arts and Science, Kasetsart University Kamphaeng Saen Campus, Nakhon Pathom 73140, Thailand; faasrrs@ku.ac.th

**Keywords:** silica, surface modification, filler, GPTMS, accelerator, rubber

## Abstract

The interaction between silica and rubber is very important for the production of high performance rubber. Silica surface modification with silane is a general method that aims to enhance the reinforcement efficiency of silica. In this study, a new surface modification of silica with silane and the chemical reaction with sulfenamide accelerator were investigated. The (gamma-glycidoxypropyl) trimethoxysilane (GPTMS) was used as a silane. The N-cyclohexyl-2-benzothiazole sulfenamide (CBS) and N-tert-butyl-2-benzothiazole sulfenamide (TBBS) were used as sulfenamide accelerators. The FTIR spectra results indicate that the GPTMS and sulfenamide accelerators (CBS and TBBS) could successfully form on the silica surface. The new modification is capable of significantly enhancing the reinforcement efficiency; more than the conventional silica surface modification by GPTMS (m-silica). In particular, modifying silica with GPTMS and TBBS (m-silica-TBBS) is capable of increasing the crosslink density and mechanical properties more efficiently than modified silica with GPTMS and CBS (m-silica-CBS), m-silica, silica (unmodified), and unfilled natural rubber. This is due to the presence of GPTMS, which plays an important role in increasing the chemical cross-linking in the rubber chain, while TBBS, as a sulfenamide accelerator, provides a high accelerator to sulfur ratio, which is able to give a more efficient vulcanization. With the reinforcement of a rubber rail pad with silica surface modification, the results indicate that the increment of m-silica-TBBS loading could reduce the deformation percentage of the rubber rail pad more than m-silica and m-silica loading. This is mainly due to the static spring improvement, which results in a stiffer material.

## 1. Introduction

Rubber rail pads are an important component of railway systems. They are interposed between steel rails and concrete sleepers, in order to protect the sleepers from wear. A rubber rail pad is an elastic material that is essential in reducing the shock and antivibration between the rail and concrete surface, due to the latter’s lack of insulation and elasticity. Therefore, rubber rail pads are widely used in many countries, to solve the insulation and elasticity problems under the rail. The optimum rubber rail pad properties are as follows: high elasticity for antivibration, high stiffness for anti-deformation, and good insulation resistance. Rubber rail pads are always made of nature rubber, due to its high tensile strength with high rebound elasticity, excellent dynamic properties, and low heat buildup. However, fillers are essential to fill in the rubber compounding, to enhance the mechanical properties of rubber composites. In general, fillers in rubber use carbon black as a traditional filler [1,2,3,4]. Several disadvantages of carbon black as a filler have been reported, including pollution, environmental problems, and waste. Many environmentally-friendly fillers have been used, such as silica [5,6,7,8], clay [9,10], calcium carbonate [11,12,13], montmorillonite [14], and halloysite [15]. These have attracted much interest for use in rubber rail pads to replace black fillers (or carbon black) because of the requirement for a low cost, as well as insulation.

Silica is a reinforcing filler that has a small particle size, high specific surface area, and avoids use of carbon [16]. Silica contributes to the enhanced stiffness of the rubber compound. Nevertheless, the silica surface is covered with silanol groups, which leads to poor dispersion of the silica in the rubber matrix and affects the cure state and mechanical properties [17,18]. This problem has to be solved to improve its reinforcing, as well as processing, characteristics. Various techniques have been proposed to improve the silica surface, such as using a silane coupling agent [19,20,21,22]. Silane coupling agents such as bis (3-triethoxysilylpropyl) tetrasulphide (TESPT) [23,24], (3-mercaptopropyl) trimethoxysilane (MPTMS) [25], bis (γ-triethoxysilylpropyl) disulfide (TESPD), and mercaptopropyltriethoxysilane (MEPTS) [6] are widely used for the modification of the silica surface in preparing rubber. The silane coupling agent can react with the silanol group on the silica surface, which improves the dispersion of silica particles in the rubber matrix [26]. The silane coupling agent can react with the silanol group on the silica surface, which is a chemical interaction between the silica surface and the rubber chain. The chemical interaction improves the dispersion of silica particles in the rubber matrix, which is beneficial to the mechanical and dynamic properties of rubber composites [27,28]. For the vulcanization of rubber, the accelerator strongly affects the crosslink density during vulcanization. The accelerator is important for high performance rubber production, and results in the stiffness of the rubber compound. The accelerator can increase the speed of vulcanization in the rubber compound [29]. The sulfenamide accelerator is the primary accelerator that it plays in good processing safety and it is a delayed action accelerator. In addition, the sulfenamide accelerator has a fast cure rate during vulcanization of rubber that it requires a long vulcanization process. Therefore, the vulcanization of rubber has a slow period of chemical reaction between the rubber chain and additives [30]. The different types of accelerators have an influence on the mechanical properties of rubber [31]. Based on the report by Nabil et al., the sulfenamide accelerator (CBS and TBBS) achieved good solubility in NR/R-EPDM blends, leading to high compatibility and strength of the blends. These accelerator groups exhibited higher tensile strengths than the other accelerators [32].

Generally, a silane coupling agent is used as a modification of the silica surface for improving silica performance. In order to enhance the performance of rubber composites, without physical loss of rubber additives, this work designed a dispersion of silica and a chemical interface between the silica and rubber with (gamma-glycidoxypropyl) trimethoxysilane (GPTMS), then chemically bonded with a sulfenamide accelerator. In this research, the (gamma-glycidoxypropyl) trimethoxysilane (GPTMS) was used to modify the silica surface (m-silica). GPTMS is a silane coupling agent that has three methoxy groups on one side and an epoxy ring on the other [33]. The methoxy groups are hydrolyzed to form a stable siloxane linkage with the silica surface. On the other side, a chemical reaction occurs, bonding with a sulfenamide accelerator such as N-cyclohexyl-2-benzothiazole sulfenamide (CBS) (known as m-silica-CBS) and N-tert-butyl-2-benzothiazole sulfenamide (TBBS) (known as m-silica-TBBS). The chemical structures of GPTMS, CBS and TBBS are presented in Figure 1. Fourier transform infrared spectroscopy (FTIR) was used to investigate the GPTMS and sulfenamide accelerator utilized on the surface of the silica. This technique has not been studied regarding rubber rail pad performance, and, therefore, this still requires further research. The effect of m-silica, m-silica-CBS, and m-silica-TBBS on the rheological characteristics, and mechanical and dynamic properties of the rubber rail pad were studied.

## 2. Materials and Methods

### 2.1. Materials

Natural rubber (STR20CV) was obtained from Thaihua Rubber, Thailand. The precipitated silica (ULTRASIL VN3) was obtained from Degussa, Thailand. GPTMS was supplied by Shin-Etsu Chemical, Japan. ZnO was obtained from Thai-Lysaght, Thailand. The N-cyclohexyl-2-benzothiazole sulfenamide (CBS), N-tertiarybutyl-2-benzothiazole sulfenamide (TBBS), diphenyl guanidine (DPG), and tetrabenzylthiuram disulphide (TBzTD) were supplied by Kawaguchi Chemical Industry, Tokyo, Japan. Other chemicals, such as stearic acid, sulfur, and N-(1,3-dimethylbutyl)-N’-phenyl-p-phenylenediamine (6PPD) were purchased from Chemmin, Thailand.

### 2.2. Surface Modification of Silica

The surface modification of silica is shown in Figure 2. The m-silica was prepared with silica surface modification by GPTMS. First, 5 g GPTMS was added to 100 mL of ethanol and stirred for 30 min. Then, 100 g of silica was added to the solution and stirred for 15 min. Finally, the m-silica was dried at 100 °C for 12 h. For the m-silica-CBS, 100 g of m-silica and 10 g of CBS were added to 100 mL of ethanol and stirred at 80 °C for 12 h. Then, the m-silica-CBS was dried at 100 °C for 12 h. The m-silica-TBBS was prepared as the m-silica-CBS with 10 g of TBBS. The surface treatment of the silica was investigated using Fourier transform infrared spectroscopy (FTIR).

### 2.3. Compounding and Vulcanization

The formulations of rubber are shown in Table 1. The compound was prepared in an internal mixer 3 L (fill factor 0.75 and rotor speed 30 rpm). The STR20CV, ZnO, and stearic acid were first masticated in a mixer for 2 min. The silica and 6PPD were added and mixed for 4 min. The accelerator and curing agent were added on a two-roll mill at room temperature (35 °C) and sheeted off at a thickness of 8 mm. Finally, the specimens were prepared by compression molding at 165 °C for 8 min.

### 2.4. Rheological Characteristics and Mechanical Properties

The rheological characteristics of the compound were measured at 170 °C for 7 min with a Moving Die Rheometer (MDR 2000, Alpha Technologies, Wilmington, DE, USA) according to ASTM 2240-93. The mechanical properties, such as hardness, tensile strength, modulus, elongation at break, and tear strength, were investigated. The tensile strength, modulus, and elongation at break of the rubber pad were investigated using a tensile testing machine (AG-IS, Shimadzu, Kyoto, Japan) at 23 ± 2 °C, with an extension speed of 500 mm/min, according to ASTM D412. The hardness was evaluated using a hardness tester (Teclock) according to ASTM D2240, with a Shore A durometer. The tear strength was measured with a type-C die according to ASTM D624.

### 2.5. Crosslink Density

A determination of the crosslink density of the rubber was obtained with a swelling test in toluene. The rubber (30 mm × 5 mm × 2 mm) was weighed and immersed in toluene at room temperature for 72 h. Then, the rubber was removed from the toluene and weighed. Swelling index was calculated with the equation [34]:(1)$$\mathrm{Swelling}\;\mathrm{index}(\%)=\frac{W_{2}-W_{1}}{W_{1}}\times 100$$
where W_1_ is the initial weight of rubber (g) and W_2_ is the weight of rubber (g) after immersion. The crosslink density of rubber was calculated using the Flory–Rehner equation [35]:(2)$$\nu =\frac{-\left[\ln\left(1-V_{r}\right)+V_{r}+\left(\chi \times V_{r}^{2}\right)\right]}{\rho_{r}\times V_{s}\times \left(\sqrt[{3}]{V_{r}}-\frac{V_{r}}{2}\right)}$$
(3)$$V_{r}=\frac{\frac{W_{1}}{\rho_{r}}}{\left(\frac{W_{1}}{\rho_{r}}\right)+\left(\frac{W_{2}-W_{1}}{\rho_{s}}\right)}$$
where ν is the crosslink density (mol/cm^3^), V_r_ is the volume fraction of rubber in the swollen gel, *χ* is the rubber–solvent interaction parameter (0.4), *ρ*_r_ is the density of rubber (0.9125 g/cm^3^), *ρ*_s_ is the density of toluene (0.867 g/cm^3^), and vs. is the molar volume of toluene (106.2 cm^3^/mol).

### 2.6. Static Spring and Deformation Testing of Rubber Rail Pad

The rubber rail pad was prepared by compression molding at 165 °C for 10 min and kept under room temperature (35 °C) for 24 h before testing. Then, the rubber rail pad was compressed with load in the range of 0–65 N. The static spring is equal to the slope of the load versus the deflection curve at the corresponding load, through the equation:(4)$$\mathrm{Static}\;\mathrm{spring}(N/\mathrm{mm})=\frac{F_{2}-F_{1}}{D_{2}-D_{1}}$$
where F_1_ is 45 N, F_2_ is 55 N, D_1_ is a displacement at 45 N, and D_2_ is a displacement at 55 N. For the deformation testing of the rubber rail pad, a load of 50 kN at 5 Hz was applied on the rubber rail pad, which was placed on a jig for around 100,000 cycles. Then, the percentage change in height of the rubber rail pad was investigated.

## 3. Results and Discussion

### 3.1. Characterization of Modified Silica

The FTIR spectra of silica (unmodified), m-silica, m-silica-CBS, and m-silica-TBBS are shown in Figure 3. The FTIR peak assignments of silica (unmodified), m-silica, m-silica-CBS, and m-silica-TBBS are shown in Table 2. The FTIR spectrum of silica (unmodified) shows a strong peak at 1090 cm^−1^ and a small peak at 799 cm^−1^, corresponding to the asymmetric stretching vibration and the symmetric stretching vibration of siloxane (Si–O–Si), respectively [36,37]. Compared to the FTIR spectra of silica (unmodified), the FTIR spectra of m-silica shows a peak at 2845 cm^−1^ and 2943 cm^−1^, attributed to the C-H stretching of CH_3_ and CH_2_. The peak at 1197 cm^−1^ is a pure vibration of methoxy groups, indicating that GPTMS was treated on the surface of silica successfully [38]. The spectrum of m-silica-CBS shows the presence of an ortho-substituted aromatic ring at 761 cm^−1^. The peaks at 1430 cm^−1^ and 1484 cm^−1^ are related to benzothiazole [39], indicating that CBS was chemically bonded onto the silica surface through the chemical linkages provided by the GPTMS. In the case of the m-silica-TBBS, the peak at 1020 cm^−1^ was observed clearly, which is due to C–N stretching, in the range of non-aromatic amines. This confirmed the achievement of chemical bonding with the TBBS accelerator onto the silica surface through the chemical linkages provided by GPTMS [40].

### 3.2. Rheological Characteristics

The cure characteristics of unfilled natural rubber and natural rubber filled with silica (unmodified), m-silica, m-silica-CBS, and m-silica-TBBS are shown in Figure 4 and Table 3. The cure curves show the vulcanization, which had an influence on the properties of the final rubber product. The optimum cure time (t_90_) is important to create a high-performance product. The optimum cure time is defined as the time necessary to obtain 90% of the maximum torque [41]. It can be seen from the results that the optimum cure time decreased significantly with modification of the silica surface by GPTMS. The results were similar to a previous report on a silane coupling agent (bis-(3-triethoxysilypropyl) tetrasulfane (Si-69) and 3-thiocyanatopropyl triethoxy silane (Si-264)) for reinforcing filler in rubber [19]. The GPTMS directly reacts with silanol groups on the silica surface, resulting in reduced accelerator adsorption. Therefore, it can be concluded that this phenomenon gives rise to a reduced curing time in the rubber compound [37]. Moreover, the influence of bonding with sulfenamide accelerator was studied. The reduction of the optimum cure time was more pronounced for m-silica-CBS than m-silica-TBBS. This may be due to an unpaired electron of a nitrogen atom from the amine group of the CBS accelerator and be ascribed to high basicity [39,42,43]. Then, the CBS accelerator cures faster than the TBBS accelerator and takes a short time to fully vulcanize. The maximum torque and the differential torque are related to the stiffness (or shear modulus) and crosslink density of the network during vulcanization [44]. The test results showed that the addition of m-silica-TBBS loading enhances the maximum torque and the differential torque, when compared to other composites, due to the ratio of accelerator to sulfur [45]. The m-silica-TBBS-filled natural rubber has a higher accelerator-to-sulfur ratio than that of m-silica-CBS. This contributes to promoting a high formation of crosslinking in rubber compounds and a stiffness enhancement. Therefore, it can be seen that the addition of m-silica-TBBS loading produced a slightly delayed reaction and a positive influence on the process safety, as well as a strongly improved crosslink density in the rubber chain; as compared to the addition of silica (unmodified), m-silica, and m-silica-CBS, respectively. Therefore, the improvement of interactions between the modified silica surface (m-silica-CBS and m-silica-TBBS) and the rubber chain are shown in Figure 5.

### 3.3. Mechanical Properties

The modified silica surface had a significant influence on the mechanical properties of the rubber. The effects of m-silica, m-silica-CBS, and m-silica-TBBS loading on the mechanical properties are summarized in Table 4. A comparison of all results clearly shows that the modification of silica surface by GPTMS increased the hardness of the rubber rail pad more than in case of silica (unmodified) and unfilled natural rubber. Consequently, the hardness values, in the order of fillers, are as follows: m-silica-TBBS > m-silica-CBS > m-silica > silica (unmodified) > unfilled natural rubber. It is well known that the maximum torque is directly correlated to hardness, corresponding to the obtained results from the rheometric curves in Figure 4. Namely, the maximum torque is increased and, thus, the hardness also tends to increase. This phenomenon also corresponds with the previous research [20]. Similarly to the hardness result, other properties such as tensile strength, tear strength, and 100% and 300% modulus were also improved with the silica surface modification by GPTMS. The reinforcing index (RI) was calculated based on the ratio of 300% modulus to 100% modulus; this indicates the reinforcing effect of the filler-filled rubber composites [46]. It can be seen that the reinforcing index of m-silica-TBBS was higher than that of m-silica-CBS, m-silica, silica (unmodified), and unfilled natural rubber (Table 4). Therefore, there was a better dispersion of m-silica-TBBS in rubber; promoting a better filler–rubber interaction in the rubber composites. The enhancement in the above properties was due to the chemical crosslinking in the rubber chain. The presence of GPTMS plays an important role in raising the chemical crosslinking in the rubber chain. This is related to the differential torque observed during vulcanization [47,48,49]. As the differential torque is increased gradually, the chemical crosslinking is also increased significantly in the natural rubber matrix. Therefore, this affects the increase of bond energy distribution and enhances the mechanical properties [50]. In addition to the chemical bonding with sulfenamide accelerator, it was also indicated that the highest value of all properties was obtained for the m-silica-TBBS loading, mainly due to the higher accelerator-to-sulfur ratio, which is able to give a more efficient vulcanization [51]. The TBBS accelerator can increase the cure state in a rubber compound more than a CBS accelerator, as confirmed by the increase of the maximum torque. By contrast, the elongation at break tends to decrease with modification of the silica surface by GPTMS; as it is widely recognized that the measure of ability to extend depends strongly on the degree of crosslinking. Therefore, GPTMS caused a high crosslink density in the rubber chain and then increased the crosslinking joints. This factor contributes to the low transfer heat that is generated by deformation through a chain motion, resulting in a breakdown at low extensions.

### 3.4. Crosslink Density

The swelling index and the crosslink density of unfilled natural rubber and natural rubber filled with silica (unmodified), m-silica, m-silica-CBS, and m-silica-TBBS are shown in Table 5. It was found that the swelling index of the rubber filled with m-silica-TBBS was lower than that of the rubber filled with m-silica-CBS, m-silica, silica (unmodified), and unfilled natural rubber. The swelling index is the amount of toluene absorption per weight of rubber, which is related to the degree of crosslink density of the vulcanized rubber. A lower value of the swelling index indicates a higher crosslink density [34]. It can be seen that the crosslink density of the rubber filled with m-silica-TBBS was higher than that of the rubber filled with m-silica-CBS, m-silica, silica (unmodified), and unfilled natural rubber. This indicated that the modification of silica with GPTMS and accelerators (CBS and TBBS) can reduce the penetration of toluene into the rubber, resulting in a relatively lower swelling index. Therefore, the penetration of toluene in rubber is restricted with m-silica-TBBS. This is due to the better dispersion of m-silica-TBBS in the rubber, promoting a better filler–rubber matrix interaction in the rubber [52]. It is well known that the crosslink density has a direct influence on the mechanical properties of rubber. It can be seen that the increase of crosslink density resulted in an increase of hardness, tensile strength, modulus, tear strength, and reinforcing index of the rubber, and a decrease of the elongation at break, due to the increasing restriction of the crosslink joints and the mobility of the inter-crosslink chains. The results can be explained in that the GPTMS and accelerators (CBS and TBBS) can improve the dispersion of silica particles and the interfacial interaction. In the report of Zhao et al. [53], the crosslink density of natural rubber was vulcanized with a different accelerator type. With the same amount of accelerator, the higher molecular weight of the accelerator resulted in a lower crosslink density of rubber. The crosslink density of DCBS (346.56 g/mol), CBS (264.40 g/mol), and TBBS (238.37 g/mol) were measured using a crosslink density spectrometer, with total crosslink density of 11.9 × 10^−5^ mol/cm^3^, 13.4 × 10^−5^ mol/cm^3^ and 14.8 × 10^−5^ mol/cm^3^, respectively [53]. With the same amount of sulfur, the crosslink density increased with the amount accelerator. Similarly, the crosslink density of the rubber filled with m-silica-TBBS was higher than that of the rubber filled with m-silica-CBS. This is due to the higher accelerator-to-sulfur ratio. Therefore, it can be concluded that the reinforcing efficiency of m-silica-TBBS is higher than that of m-silica-CBS.

### 3.5. The Static Spring and Deformation Testing of Rubber Rail Pad

The silica surface was modified using GPTMS and filled into natural rubber to produce a rubber rail pad. Load versus displacement graphs of the rubber rail pad are shown in Figure 6. It can be seen that the displacement of the rubber rail pads with modified silica (m-silica, m-silica-CBS, and m-silica-TBBS) decreased gradually (as load at 45 N or 55 N), as compared to filled silica (unmodified) and unfilled natural rubber. This is due to the high interaction between the silica and natural rubber, resulting in an inability to distort the shape of the rubber rail pad. For the m-silica-TBBS loading, the displacement was also decreased more than m-silica-CBS, m-silica, and silica (unmodified), respectively. This was due to the enhancement of the stiffness of the rubber rail pad. The enhanced stiffness value provides a very strong and rigid structure. This is defined as the ability of the rubber rail pad to resist the elastic segment to deflection by an applied load without permanent deformation. It is important to note that the static spring of rubber rail pad was calculated from Figure 6 (equal to the slope of the load versus displacement curve) and summarized in Table 6. The static spring can be largely explained by the rubber rail pad’s performance, such as a higher static spring meaning a stiffer material. The static spring value is also related to the crosslink density in the rubber chain. A high static spring value indicates a high crosslink density in the rubber chain, which agreed with the hardness results (Table 4) [23]. It is well known that the stiffness is closely related to the hardness of a material. The static spring of the rubber rail pad filled with m-silica-TBBS (588.235 N/mm) increased by about 5.55%, 22.73%, 41.38%, and 83.80% compared to filled m-silica-CBS (555.556 N/mm), m-silica (454.545 N/mm), silica (unmodified) (344.828 N/mm), and unfilled natural rubber (95.238 N/mm), respectively. The test results indicate that m-silica-TBBS loading contributes to a rigid rubber rail pad more than m-silica-CBS, m-silica, and silica (unmodified). Further analysis of the deformation testing was performed, to measure the percentage change in height of the rubber rail pad when subjected to repeated loading and unloading. The deformation percentage of the rubber rail pad is illustrated in Figure 7. The results can be ordered from high to low, as follows: unfilled natural rubber > silica (unmodified) > m-silica > m-silica-CBS > m-silica-TBBS. An improvement in the best deformation resistance was observed for the rubber rail pad filled with m-silica-TBBS. The rubber rail pad filled with m-silica-TBBS showed the lowest deformation percentage of 3.77%, which is a good for withstanding deformation, and it returned to its original shape after unloading. This was achieved due to a greater crosslink density in the rubber chain. As the crosslink density increases, the rubber molecules can be strengthened, and without breaking bonds when compressed. Therefore, a higher applied load is needed to break the bonds and increase the deformability of the rubber rail pad. The unfilled and silica (unmodified) filled rubber rail pad were easily deformed during usage, about 23.29% and 15.28%, respectively. This is attributed to there being no filler in the rubber matrix and the weaker linkage taking place at the rubber chain, due to its unmodified surface. Weak chemical bonds can cause rubber rail pad deterioration. As shown by the test results, deform resistance not only depends on the degree of crosslinking in the rubber chain, but also the static spring, which helps to prevent failure of the rubber rail pad. This result was correlated by the successful surface modification of silica by the GPTMS and TBBS accelerators, more than the conventional silica surface modification by GPTMS only. TBBS are chemically bonded onto the surface of silica in the structure of m-silica. Thus, the rubber chains can directly vulcanize on the surface of the silica and entangle themselves with GPTMS, which is chemically bonded with TBBS, as revealed by the FTIR and Figure 2. Consequently, the surface of the m-silica-TBBS had a crosslinked network of rubber chains after vulcanization. The formation of stretched polymer chains dominates the reinforcing efficiency of the m-silica-TBBS. Therefore, it is suggested that modified silica with GPTMS and TBBS accelerators could be used as an alternative filler in rubber rail pad production.

## 4. Conclusions

A silica surface was modified using silane, then chemically bonded with sulfenamide accelerators. GMPTS and sulfenamide accelerators (CBS and TBBS) were successfully treated onto the silica surface, as confirmed by FTIR. The silica surface modification was capable of significantly improving the reinforcement efficiency of natural rubber composites. The m-silica-TBBS enhanced the mechanical properties, especially, hardness, tensile strength, tear strength, 100% and 300% modulus, more than m-silica-CBS, m-silica, silica (unmodified), and unfilled natural rubber. This is due to the high accelerator-to-sulfur ratio, leading to a better efficiency of vulcanization and an enhancement of crosslink density. In the case of the rheological characteristics, the incorporation of m-silica-TBBS enhanced the stiffness or shear modulus, and also reduced the curing time, because it reduced the accelerator adsorption. The crosslink density of m-silica-TBBS (11.83 × 10^−5^ mol/cm^3^) was higher than that of m-silica-CBS (10.92 × 10^−5^ mol/cm^3^), m-silica (9.66 × 10^−5^ mol/cm^3^), and silica (unmodified, 8.16 × 10^−5^ mol/cm^3^). The increase of crosslink density resulted in an increase of hardness, tensile strength, modulus, tear strength, and reinforcing index of the rubber, and a decrease of the elongation at break, due to the increase in the restriction of the crosslink joints and the mobility of intercrosslink chains. Additionally, the reinforcement of the rubber rail pad with m-silica-TBBS, m-silica-CBS, m-silica, and silica (unmodified) for the static spring and deformation properties were also evaluated. The results suggested that the presence of GMPTS and TBBS on the silica surface increased the effectiveness of silica as a filler in the rubber rail pad. This was due to an increase in the static spring and resulted in reducing the deformation percentage of the rubber rail pad with m-silica-TBBS more than m-silica-CBS, m-silica, silica (unmodified), and unfilled natural rubber. Therefore, the reinforcement efficiency of the m-silica-TBBS was higher than that of m-silica-CBS, m-silica, and silica (unmodified). The better dispersion of m-silica-TBBS in rubber promoted a better filler–rubber interaction in the rubber composites. This is due to the high crosslink density in the rubber chain and higher accelerator-to-sulfur ratio. This new surface modification of the silica surface with GPTMS and TBBS accelerator could be an alternative reinforcing filler for the preparation of high-performance rubber composites.

## Figures and Tables

**Figure 1 polymers-14-01767-f001:**
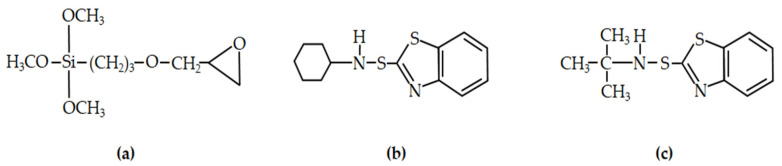
Chemical structures of (**a**) GPTMS, (**b**) CBS, and (**c**) TBBS.

**Figure 2 polymers-14-01767-f002:**
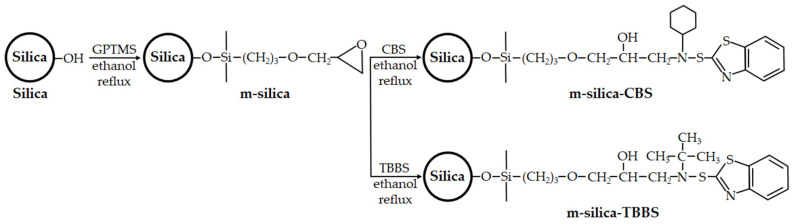
Synthesis of m-silica, m-silica-CBS, and m-silica-TBBS.

**Figure 3 polymers-14-01767-f003:**
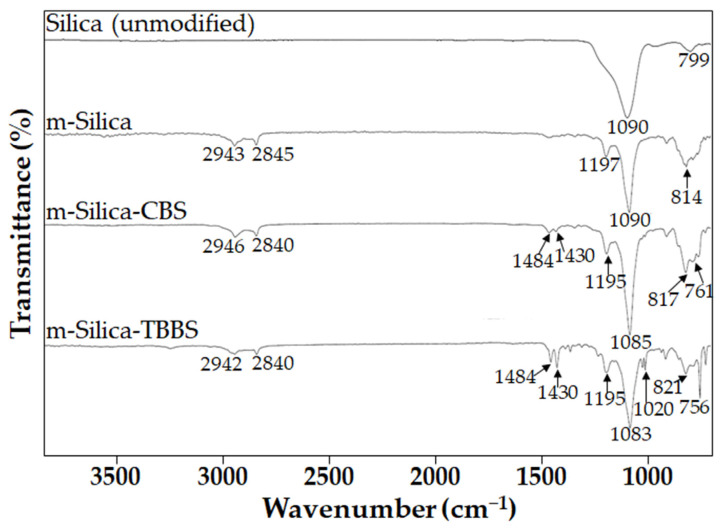
FTIR spectra of silica (unmodified), m-silica, m-silica-CBS, and m-silica-TBBS.

**Figure 4 polymers-14-01767-f004:**
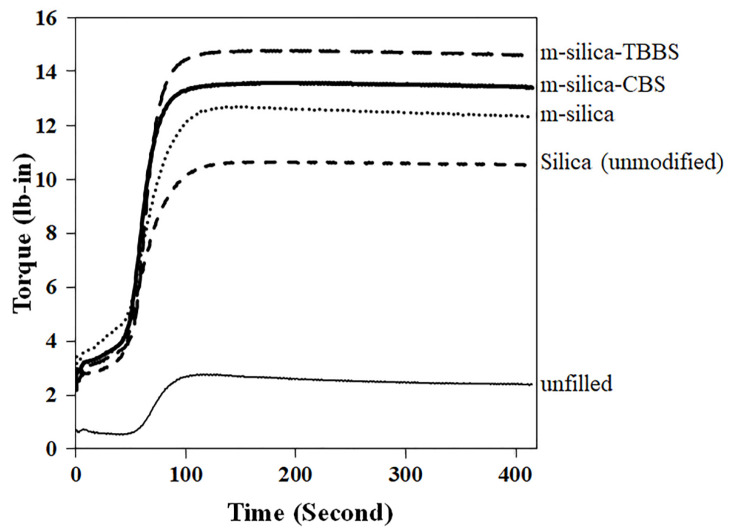
Cure curves of unfilled natural rubber and natural rubber filled with silica (unmodified), m-silica, m-silica-CBS, and m-silica-TBBS.

**Figure 5 polymers-14-01767-f005:**
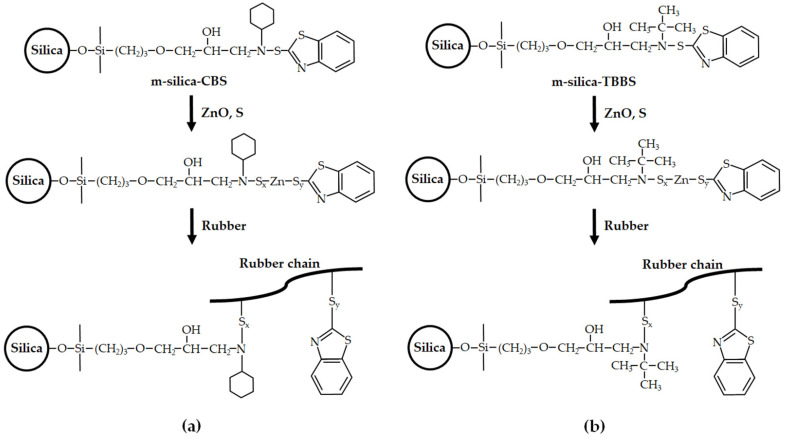
Chemical reactions of the modified silica surface in rubber (**a**) m-silica-CBS and (**b**) m-silica-TBBS).

**Figure 6 polymers-14-01767-f006:**
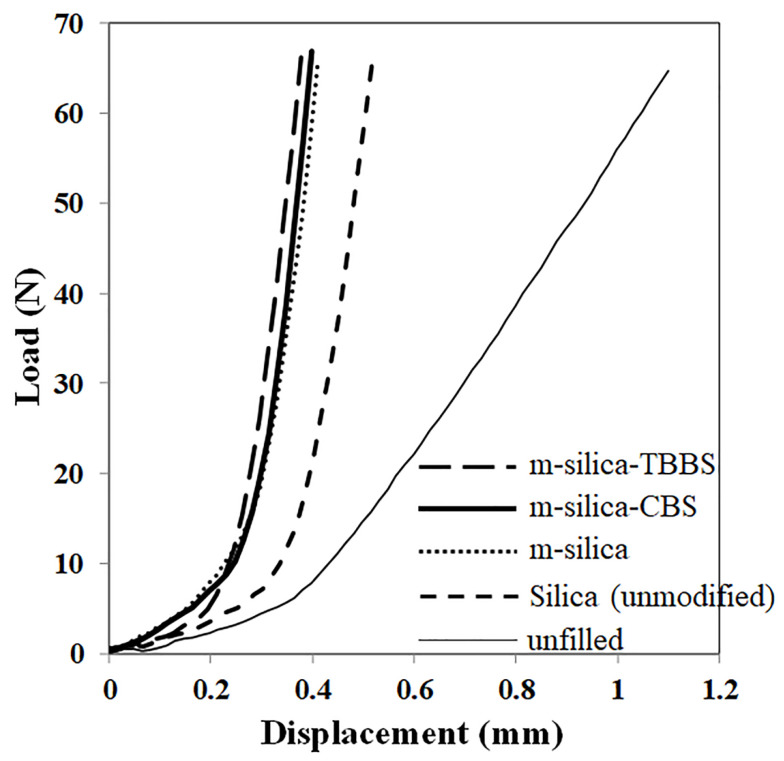
Load versus displacement graphs of rubber rail pads.

**Figure 7 polymers-14-01767-f007:**
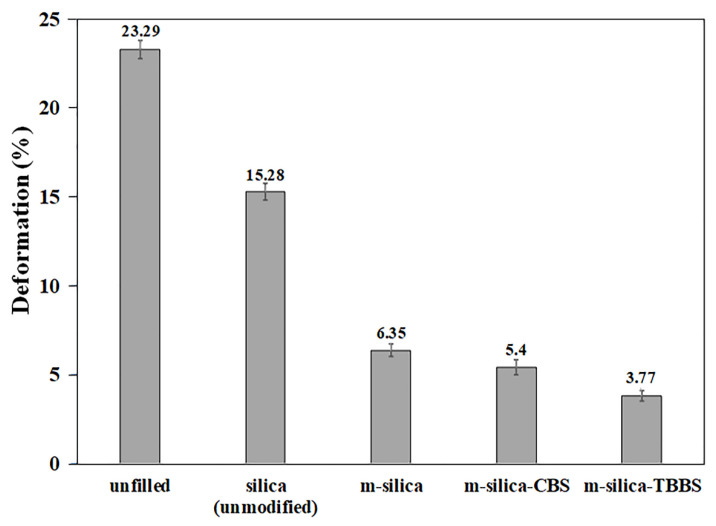
The deformation percentage of rubber rail pads.

**Table 1 polymers-14-01767-t001:** Formulations of rubber.

Materials (phr *)	Unfilled	Silica (Unmodified)	m-silica	m-silica-CBS	m-silica-TBBS
STR 20CV	100	100	100	100	100
N-774	-	35	35	35	35
Silica	-	35	-	-	-
m-silica	-	-	35	-	-
m-silica-CBS	-	-	-	35	-
m-silica-TBBS	-	-	-	-	35
ZnO	5	5	5	5	5
Stearic acid	1	1	1	1	1
6PPD	1	1	1	1	1
TBzTD	1	1	1	1	1
DPG	0.5	0.5	0.5	0.5	0.5
Sulphur	2	2	2	2	2

* phr (parts per hundred parts of rubber).

**Table 2 polymers-14-01767-t002:** FTIR peak assignments of silica (unmodified), m-silica, m-silica-CBS, and m-silica-TBBS.

FTIR Peak (cm^−1^)				Assignment
Silica (Unmodified)	m-silica	m-silica-CBS	m-silica-TBBS	
-	-	761	-	Ortho-substituted aromatic ring
799	814	817	821	Symmetric stretching vibration of siloxane
-	-	-	1020	C–N stretching
1090	1090	1085	1083	Asymmetric stretching vibration of siloxane
-	1197	1195	1195	Pure vibration of methoxy groups
-	-	14,301,484	14,301,484	Benzothiazole
-	28,452,943	28,402,946	28,402,942	C–H stretching of CH_3_ and CH_2_

**Table 3 polymers-14-01767-t003:** Cure characteristics of unfilled natural rubber and natural rubber filled with silica (unmodified), m-silica, m-silica-CBS, and m-silica-TBBS.

Properties	Unfilled	Silica (Unmodified)	m-silica	m-silica-CBS	m-silica-TBBS
Maxmimum torque (M_H_), lb-in	2.74	10.64	12.69	13.59	14.79
Minimum torque (M_L_), lb-in	0.49	2.53	3.26	2.83	2.48
Differential torque (M_H_-M_L_), lb-in	2.25	8.11	9.43	10.76	12.31
Optimum cure time (t_90_), second	2.52	9.83	11.75	12.51	13.56

**Table 4 polymers-14-01767-t004:** The mechanical properties of unfilled natural rubber and natural rubber filled with silica (unmodified), m-silica, m-silica-CBS, and m-silica-TBBS.

Properties	Unfilled	Silica (Unmodified)	m-silica	m-silica-CBS	m-silica-TBBS
Hardness (Shore A)	26 ± 1	57 ± 1	58 ± 1	61 ± 1	63 ± 1
Tensile strength (MPa)	11.18 ± 0.22	15.49 ± 0.32	17.77 ± 0.18	19.88 ± 0.20	23.36 ± 0.18
Elongation at break (%)	940 ± 20	580 ± 30	560 ± 30	540 ± 20	525 ± 20
100% Modulus (MPa)	1.32 ± 0.02	1.70 ± 0.04	1.98 ± 0.02	2.42 ± 0.01	2.72 ± 0.02
300% Modulus (MPa)	2.38 ± 0.03	5.76 ± 0.02	7.91 ± 0.05	10.83 ± 0.03	12.27 ± 0.03
Tear strength (kN/m)	19.77 ± 0.05	50.45 ± 0.02	56.11 ± 0.03	61.54 ± 0.04	71.08 ± 0.02
Reinforcing index (RI)	1.80	3.39	3.99	4.48	4.51

**Table 5 polymers-14-01767-t005:** The swelling index and the crosslink density of unfilled natural rubber and natural rubber filled with silica (unmodified), m-silica, m-silica-CBS, and m-silica-TBBS.

Properties	Unfilled	Silica (Unmodified)	m-silica	m-silica-CBS	m-silica-TBBS
Swelling index (%)	572	531	485	454	435
Crosslink density (mol/cm^3^)	7.11 × 10^−5^	8.16 × 10^−5^	9.66 × 10^−5^	10.92 × 10^−5^	11.83 × 10^−5^

**Table 6 polymers-14-01767-t006:** The displacement and the static spring of rubber rail pads.

Formulation	Displacement (mm)		Static Spring (N/mm)
	Load at 45 N	Load at 55 N	
Unfilled	0.865	0.970	95.238
Silica (unmodified)	0.494	0.523	344.828
m-silica	0.361	0.383	454.545
m-silica-CBS	0.360	0.378	555.556
m-silica-TBBS	0.347	0.364	588.235

## Data Availability

The data presented in this study are available on request from the corresponding author.

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
