# Peer review of "Rubber Rail Pad Reinforced by Modified Silica Using GPTMS and Sulfenamide Accelerator"

_polymers, 2022, doi:10.3390/polym14091767_

Round 1

Reviewer 1 Report

In this work, the authors reported on the rubber rail pads that are essential in reducing shock and antivibration between rail and concrete surface, due to the latter’s lack of insulation and elasticity.  For this, the interaction between silica and rubber is important for the production of high performance rubber. Usually, silica surface modification by silane is a general method which aims to enhance the reinforcement efficiency of silica. In this study, the authors modified surface of silica by silane and the chemical reaction with sulfenamide accelerator. They found that modified-silica with GPTMS and TBBS (m-silica-TBBS) increased the crosslink density and mechanical properties more efficiently than modified-silica with GPTMS and CBS (m-silica-CBS), m-silica, silica (unmodified) and unfilled natural rubber, respectively.  They also found in the rubber rail pad with silica surface modification, the increment of m-silica- 23 TBBS loading can reduce the deformation percentage of rubber rail pad more than m-silica and m- 24 silica loading due to the static spring improvement. The conclusions are sound and is worthy of publication in Polymers after major revision.

My specific comments are;

  1. Though they reported the crosslinking degree by torque changes, more desirables are to report crosslinking density more scientifically, for instance, using gel contents experimentally and theorectically by Flory theory.
  2. Distribution of silica in the rubber should be verified using SEM with EDX, etc.

Reviewer 2 Report

In order to improve the performance of rubber composites used in rubber rail pads, this work is studying dispersion of silica and chemical interface between silica and rubber with (gamma- 63 glycidoxypropyl) trimethoxysilane (GPTMS) and sulfenamide accelerator.

The paper is well designed and the presentation is cursive and clear. Description of the results can be improved as follows:

- FTIR analysis is well explained, but probably discussion will be clearer if a Table summarizing the main bands will be provided (in chapter 3.1).

- If vulcanization parameters (scorch time, vulcanization time, minimum torque, torque increment) were entered in a table, they would be much easier to identify and track - I am referring to chapter 3.2. Rheological characteristics.

- If the mechanical results were represented graphically, they would be much more suggestive and obvious (in chapter 3.3).

- As a suggestion: try to determine the gel fraction and the degree of crosslinking. The working method is found in the literature and is not complicated. However, these results would add value to the article. If you do not succeed in this work, keep in mind for the future.

- The conclusions should be supported by the most representative experimental results.

Reviewer 3 Report

What I really like about this work is its high applicability. The author focused his attention on a widely used polymer and designed-directed its modification aiming to solve a real issue of an essential and well-needed benefit, railway systems. The author demonstrated that by using surface-modified silica as filler in rubber-based formulation, it is possible to increase the mechanical properties of the final material, allowing it to fulfill in a better way the requirements for its use as rail pads. One of the novelties of the present work is the use of CBS and TBBS as entities able to act as accelerators in the process. However, I am convinced that the scientific soundness, along with the characterization of the final materials, must be improved before being considered for publication in this journal.

Here are some points I would like to know the author´s opinion:

1) Why did you choose CBS and TBBS? What was the real motivation to use these structures? are they interesting from an economic perspective? (Also, fix the CBC draw in Figure 1, cyclohexane appears to be distorted)

2) What are the units of the numbers in Table 1?

3) From FTIR analysis, it seems that a higher surface modification is achieved by using TBBS. The above could be related to the higher bulkiness of the cyclohexyl group. Nevertheless, I believe that an elemental analysis of samples should be carried out in order to get an accurate image of this situation (Based on the N %, for example). If we want to compare fillers modified with different agents, it is mandatory to have a similar surface modification percentage. From your results, without being sure of the degree of modification, it is not possible to demonstrate which agent is really the best. Maybe the better performance exhibited by TBBS could be ascribed mainly to a higher presence on the silica's surface rather than a chemical behavior. I insist the author must be solve this issue, since all the arguments and discussions given in the present manuscript are highly dependent of this value

4) Lines 150-151: "The optimum cure time is defined as the time necessary to obtain 90% of the maximum torque" Can you give references for this statement?

5) By last, I would appreciate a scheme showing the crosslinking mechanism expected in these samples.

Round 2

Reviewer 1 Report

It seems to me that the authors incorporated my comments except the crosslink density. Though the authors added the crosslink density values in the text, there are no description how they obtained the crosslink density values. Did they obtain the values experimentally ? Then how? If they obtained that values theoretically and then how? More detailed description is needed on the crosslink density.

Reviewer 3 Report

The author solved most of the suggestions

Round 3

Reviewer 1 Report

Though they tried to improve the manuscript according to my comments, there are neither experimental methods to check the crosslinking density (though they supplied the values reported values) nor TEM or EDX results (though they described something just with a few sentences). Thus those experimental data should be supplemented in their revised version. 

Round 4

Reviewer 1 Report

Now the manuscript has been much improved.